# Therapeutic Effects of Mesenchymal Stromal Cells Require Mitochondrial Transfer and Quality Control

**DOI:** 10.3390/ijms242115788

**Published:** 2023-10-31

**Authors:** Avinash Naraiah Mukkala, Mirjana Jerkic, Zahra Khan, Katalin Szaszi, Andras Kapus, Ori Rotstein

**Affiliations:** 1Unity Health Toronto, The Keenan Research Centre for Biomedical Science of St. Michael’s Hospital, University of Toronto, Toronto, ON M5B 1T8, Canada; avinash.mukkala@mail.utoronto.ca (A.N.M.); zahra.khan@mail.utoronto.ca (Z.K.); katalin.szaszi@unityhealth.to (K.S.); andras.kapus@unityhealth.to (A.K.); ori.rotstein@unityhealth.to (O.R.); 2Institute of Medical Science, University of Toronto, Toronto, ON M5S 1A8, Canada; 3Department of Surgery, University of Toronto, Toronto, ON M5T 1P5, Canada

**Keywords:** mitochondria, mitochondrial transfer, microvesicles, cell treatment, injury resolution

## Abstract

Due to their beneficial effects in an array of diseases, Mesenchymal Stromal Cells (MSCs) have been the focus of intense preclinical research and clinical implementation for decades. MSCs have multilineage differentiation capacity, support hematopoiesis, secrete pro-regenerative factors and exert immunoregulatory functions promoting homeostasis and the resolution of injury/inflammation. The main effects of MSCs include modulation of immune cells (macrophages, neutrophils, and lymphocytes), secretion of antimicrobial peptides, and transfer of mitochondria (Mt) to injured cells. These actions can be enhanced by priming (i.e., licensing) MSCs prior to exposure to deleterious microenvironments. Preclinical evidence suggests that MSCs can exert therapeutic effects in a variety of pathological states, including cardiac, respiratory, hepatic, renal, and neurological diseases. One of the key emerging beneficial actions of MSCs is the improvement of mitochondrial functions in the injured tissues by enhancing mitochondrial quality control (MQC). Recent advances in the understanding of cellular MQC, including mitochondrial biogenesis, mitophagy, fission, and fusion, helped uncover how MSCs enhance these processes. Specifically, MSCs have been suggested to regulate peroxisome proliferator-activated receptor-gamma coactivator 1 alpha (PGC1α)-dependent biogenesis, Parkin-dependent mitophagy, and Mitofusins (Mfn1/2) or Dynamin Related Protein-1 (Drp1)-mediated fission/fusion. In addition, previous studies also verified mitochondrial transfer from MSCs through tunneling nanotubes and via microvesicular transport. Combined, these effects improve mitochondrial functions, thereby contributing to the resolution of injury and inflammation. Thus, uncovering how MSCs affect MQC opens new therapeutic avenues for organ injury, and the transplantation of MSC-derived mitochondria to injured tissues might represent an attractive new therapeutic approach.

## 1. Mesenchymal Stromal Cells (MSCs) in Health and Disease

### 1.1. Definition and Characteristics of MSCs

MSCs are multipotent, non-hematopoietic cells capable of differentiating into a variety of mature cell types including adipocytes, chondrocytes, osteoblasts, and myoblasts [1]. In the literature, both terms, mesenchymal stem and mesenchymal stromal cells, can be found [2], and sometimes they are interchangeably used. Clarification of the terms is issued by the International Society for Cell and Gene Therapy (the ISCTs MSC committee criteria) [3], defining mesenchymal stem cell populations as progenitor cells capable of self-renewal and differentiation. Multipotent stromal cells (MSCs) are plastic-adherent fibroblast-like cells that express CD73, CD90 and CD105, and lack the expression of hematopoietic and endothelial markers CD11b, CD14, CD19, CD34, CD45, CD79a, and the human leukocyte antigen (HLA)-DR. These cells support hematopoiesis and have strong immunomodulatory properties, due to the secretion of an array of growth factors, cytokines, and the production of extracellular vesicles that contain mitochondria that can be transferred to recipient cells. MSCs also have homing properties. Due to the absence or negligible expression of HLA-DR, MSCs are considered immuno-privileged cells, allowing for their allogeneic use. In addition, their relative simplicity of isolation and culture, as well as the abovementioned MSC characteristics, have encouraged their use in hundreds of clinical trials. Overall, in the field of cell-based therapies, MSCs represent the most promising option [4].

### 1.2. MSC Therapeutic Use in Various Disorders

MSCs have the ability to promote the repair of damaged tissue, and they have been shown to be beneficial in a variety of disorders, including sepsis [5], lung diseases [6], liver diseases [7], ischemia/reperfusion (I/R) injury [8], and osteoarthritis [9]. It is important to emphasize that MSC treatment is not always favorable, as MSCs can cause adverse reactions or even detrimental effects, depending on the cell secretory and genetic profile, microenvironment composition, and time of the cell administration. A preclinical study by Islam et al. [10] pointed out that MSC treatment was favorable for ventilator-induced lung injury, but detrimental for acid-induced lung injury, resulting in lung fibrosis. These harmful MSC effects were caused by the microenvironment containing high levels of IL-6 and fibronectin, accompanied by low total antioxidant capacity. Therefore, strategies to prevent the unfavorable effects of MSCs could include modification of such environments, specific biomarker measurements in patients, in vitro microenvironment simulation, and gene-expression modulation in MSCs before treatment, along with a personalized approach for MSC use in specific conditions [10,11,12,13,14].

MSCs have shown beneficial effects in a variety of preclinical animal models, and they have been used widely in human clinical trials. However, there are a number of important limitations in the available data that precludes moving towards broad clinical use. Importantly, so far, clinical trials have been mostly restricted to the examination of safety and efficacy of MSCs from different sources, the exploration of various modes of administration, or the use of MSC-conditioned media containing secreted factors, extracellular vesicles (EVs), and mitochondria (Mt). In addition, MSCs were mostly administered in patients with severe diseases when standard treatment resulted in no improvement. Thus, information on possible benefits in earlier disease stages is mostly unavailable.

The most widely used MSC types are allogeneic bone marrow (BM) MSCs, umbilical cord (UC) MSCs, or MSC-derived extracellular vesicles (MSC-EVs).

MSCs are being tested in hundreds of clinical trials [15] in critically ill patients with sepsis [16], for the prevention of graft vs. host disease (GVHD) [14], lung and liver carcinoma [17], in acute respiratory distress syndrome (ARDS) [18], COVID-19 pneumonia [19], and for benefits in wound healing [20].

The mechanisms of action of MSC or MSC-EV delivered to injured tissues were explored mainly in preclinical studies [21]. These studies demonstrated several key effects, including various immunomodulatory actions of MSCs through cell–cell contact and/or paracrine and autocrine activities. These latter effects include the delivery of cargo consisting of Mt, mRNA, miRNA, cytokines, and growth factors to target cells.

A key immunomodulatory effect of MSCs is the enhancement of bacterial killing. MSCs and MSC-EVs increase macrophage (Mϕ) phagocytic capabilities and skew Mϕ polarization toward an M2-like anti-inflammatory phenotype. They supress neutrophil infiltration, and regulate dendritic cell maturation and T lymphocyte proliferation, as well as B lymphocyte antibody secretion [1]. MSCs also secrete antimicrobial peptides, including defensins, LL-37, and indoleamine [22], release EVs, and act through organelle transfer to donor cells.

Crucially, a multitude of studies have revealed that one of the key outcomes of MSCs treatment is the improvement of Mt homeostasis in injured tissues and immune cells, due to enhanced MQC processes. One underlying mechanism is the donation of healthy Mt from the MSCs to cells. Accordingly, recent studies have demonstrated that the ability of MSCs to act as effective mitochondrial donors is crucial for their therapeutic efficacy [23,24,25,26,27,28].

MSC efficacy can be augmented by various priming or “licensing” strategies that take advantage of a key feature of MSCs, namely, their potential to respond to microenvironment change [13]. By pre-exposing MSCs to injury-associated cues, such as hypoxia or cytokines that are found in the damaged tissue, MSCs become activated and prepared to act on the injured tissues [29]. Notably, such activated MSCs showed increased Mt transfer to recipient cells [30] and were able to mitigate oxidative stress in acute lymphoblastic leukemia (ALL) cells and in a murine ALL model. The importance of Mt was also demonstrated in these activated MSCs, as depleting Mt or preventing their transfer abolished the rescue function of activated MSCs [30]. In fact, the therapeutic efficacy of MSCs appears to depend on the functional characteristics of their mitochondria (Mt) and the capacity to donate Mt as well as the regulation of mitochondrial uptake by recipient cells [31]. For example, enhancement of MSC-induced Mϕ phagocytic activity is partly dependent on the uptake of MSC-EVs containing Mt [32].

Taken together, Mt biology plays a central role in the therapeutic efficacy of MSCs. First, they are highly reliant on efficient MQC for their own functions. Second, the ability of MSCs to donate and transfer Mt, and to improve MQC in recipient cells, appears to be vital for their therapeutic effects on injured organs. Therefore, in the next section we will provide an overview of the process of cellular MQC and its role in healthy cells and benefits in promoting the recovery of damaged cells

## 2. Mitochondria (Mt) and Their Role in Health and Disease

### 2.1. Physiological Importance and Function of Mitochondria

Mt are the main generators of cellular chemical energy, and they are essential for the maintenance of cellular health [33]. Mitochondrial structure is dynamic and under constant change while aiming at meeting specific cellular demands. The mitochondrial network and its contact sites with other organelles, especially the endoplasmic reticulum, can provide a structural pathway for energy distribution and communication across long cellular distances. An efficient quality-control system ensures that the mitochondrial network can fulfill its vital functions and meet changing needs.

As discussed above, healthy mitochondria are essential for the efficient function of MSCs, and emerging research has pointed to the vital role of improved mitochondrial functions in injured tissues as a key mechanism whereby MSCs exert their pro-healing effects. While there are large gaps in our understanding of how this happens, elements of the MQC process appear to be essential for these effects.

### 2.2. Mitochondrial Quality Control (MQC)

#### 2.2.1. Mitochondrial Fission/Fusion and Their Regulation

Mt are dynamic organelles that undergo fission and fusion (Figure 1), oscillating between a particulate/globular and a tubular/elongated morphology [34,35]. Importantly, the fission–fusion balance both regulates, and is regulated by, metabolic/oxidative states and a variety of associated mitochondrial functions, including the control of cell death/survival and differentiation [36]. Fission and fusion have key physiological functions, and their proper, context-dependent balance is necessary for the structural and functional integrity of the mitochondrial network and the cell as a whole (Figure 1). Mitochondrial fission is essential for the proper distribution of Mt into the daughter cells during division [37,38]; it limits the spread of mitochondrial dysfunction (such as local Ca2+ overload) to subcompartments, thereby protecting intact mitochondria [39], and serves mitophagy by generating “edible” smaller units [40,41]—an essential component of quality control. On the other hand, excessive fission is both a major sign and an inducing factor of cell stress, and is associated with deterioration of ATP production, extensive ROS generation, apoptosis, or necrotic death [37,42]. Mitochondrial fusion ensures the coordinated regulation and operation of the entire mitochondrial network (e.g., uniformly high membrane potential); the sharing of intra-mitochondrial material (e.g., mtDNA) thereby compensating for local losses; and is usually associated with, and necessary for, efficient oxidative phosphorylation (OXPHOS) and cristae formation [43]. However, excessive fusion (hyperfusion) promotes cellular senescence [44], disrupts normal mitochondrial energy production, and interferes with mitophagy [45,46,47]. Genetic alterations of either the fission or the fusion machinery lead to inherited mitochondrial diseases, including motor and sensory neuropathies, optic nerve atrophy, myopathies, and various developmental defects [48,49].

Here, we briefly summarize key proteins constituting the fission and fusion apparatus, and then highlight some aspects of mitochondrial dynamics, which are specifically relevant to the biology of stem cells in general or MSCs in particular. For a more detailed exposition of the latter topics, the reader is referred to recent reviews [50,51,52,53,54].

Fission (fragmentation) and fusion of mitochondrial membranes are catalyzed by large mechano-enzymes, belonging to the family of Dynamin-like GTPases, and is aided by receptors/accessory proteins involved in their recruitment or regulation. The key component of mitochondrial fission is Dynamin-related protein-1 (Drp1), which, upon induction of fragmentation, can translocate from the cytosol to the outer mitochondrial membrane (OMM). There it forms a ring-like polymer, which surrounds and squeezes the mitochondrion, ultimately separating it into smaller units [36,55]. The receptor for Drp1 on the OMM is Fis1, while accessory proteins such as mitochondrial fission factor (Mff) and mitochondrial dynamics 49 and 51 (MiD49 and MiD51) may facilitate the Drp1/Fis1 interaction and Drp1 translocation [56,57,58]. Drp1 localization and activity is primarily regulated by its posttranslational modification, including both activating and inhibitory phosphorylation by a variety of kinases [53,59,60], as well as ubiquitination and sumoylation [61,62,63]. Moreover, Drp1 translocation is facilitated by cytoskeletal changes, mediated by inverted formin 2 (INF2) and local myosin activation. These proteins preferentially elevate Drp1 levels and sites of mitochondrial contraction that occur at the mitochondrion–endoplasmic reticulum contacts (MERCs) [64,65,66]. MERCs are also specific loci for direct transfer of Ca2+ from the ER to mitochondria [67,68]. High intra-mitochondrial Ca2+ can in turn promote fission [69], linking mitochondrial ion transport to dynamics. Fragmentation is provoked by many cellular stress conditions relevant to organ injury, e.g., by poisons or ischemia-reperfusion injury [37,42,70,71,72]. As mentioned, the process can initially serve restorative/compensatory purposes (segregation, autophagy), but excessive fragmentation leads to dramatic deterioration of OXPHOS and can trigger cell death [37]. Further, while fission (or an appropriately small size) is necessary for engulfment during autophagy, the role of Drp1 in mitophagy is controversial [40]. In this regard, the recent discovery of Atg44 (named mitofissin) is of great interest. This protein induces Dnm1L (yeast homolog of Drp1)-independent fission, and is indispensable for mitophagy [41,73]. Identification of the mammalian homologs/equivalents of Atg44 remains an exciting future goal.

The opposite process, fusion, is catalyzed by the transmembrane GTPases, mitofusin 1 and 2 (Mfn1 and Mfn2) in the OMM and optic atrophy 1 (OPA1) in the inner mitochondrial membrane (IMM) [74,75,76]. Mitofusins also localize to MERCS, and Mnf2 emerged as a regulator of mitochondria–ER tethering, albeit both positive and negative roles were reported [77,78]. The accessory protein Misato was shown to promote the activity of Mfns [79]. OPA1, which mediates IMM fusion after Mfn-induced OMM fusion, can be cleaved by the protease OMA in the intermembrane space. The resulting short OPA1 (S-OPA) is an inhibitor of fusion. Interestingly, bioenergetic intactness, i.e., high membrane potential, is a central regulator of these processes: it allows OPA1 to carry out fusion, whereas the loss of membrane potential facilitates the cleavage of OPA1 [80].

The ubiquitin kinase PINK1 and the ubiquitin ligase Parkin1, central mediators of autophagy, have also emerged as regulators of the fusion/fission balance; however, their overall effect is controversial. A subset of studies proposed that the knockdown of these proteins promoted mitochondrial fragmentation [81,82,83], thus they are pro-fusion (as viewed in Seo et al., 2018) [52]. However, other reports claimed the opposite, showing that their elimination leads to net fusion [84,85,86] (and see early Drosophila studies reviewed in Ge et al., 2020) [87]. The potential explanation may lie in the fact that PINK1/Parkin1 specifically augment the elimination of defective, depolarized mitochondria [88,89], whereas they might stabilize functionally intact mitochondria. Parkin may promote the ubiquitination of both Drp1 [90] and Mfn [91]. In any case, this scenario exemplifies a fine-tuned interplay between the regulation of fusion and fission, and shows the relationship between mitochondrial dynamics and QC.

#### 2.2.2. Mitophagy and Mitochondrial Biogenesis

The maintenance of mitochondrial network integrity is orchestrated by the processes of mitophagy and mitochondrial biogenesis (Figure 1). Autophagy of Mt, known as mitophagy, is a cellular mechanism by which dysfunctional mitochondria are cleared from the mitochondrial network. In this manner, mitophagy is crucial in the restoration of mitochondria homeostasis. Mitochondrial biogenesis is the synthesis of new mitochondria. Replication of mitochondria DNA, transcription of mitochondrial genes (encoded both by the nuclear and mitochondrial genomes), and mitochondrial protein synthesis all contribute strongly to the increase of total cellular mitochondria numbers. Together, mitochondrial biogenesis and mitophagy have strong effects on metabolism, on bioenergetics, and therefore on cellular homeostasis [92].

The key mitophagy pathways (Figure 1) are divided into receptor-mediated mitophagy and ubiquitin-mediated mitophagy [93,94]. During receptor-mediated mitophagy, under mitochondrial stress (i.e., hypoxia or depolarizing agents), mitophagy receptors localize to the outer mitochondrial membrane. Two of the mammalian mitophagy receptors include FUN14 domain-containing 1 (FUNDC1) and BCL2/adenovirus E1B 19 kDa protein-interacting protein 3-like (BNIP3L)/NIP3-like-protein X (NIX) [95,96], both of which contain conserved LC3-binding motifs to facilitate phagophore formation. Lysosomal fusion with mitophagophores facilitates degradation. In ubiquitin-dependent mitophagy, the best understood pathway is PINK1/Parkin-dependent mitophagy [97,98,99]. Under mitochondria stress, PINK1 (a serine/threonine kinase) and Parkin (an E3 ubiquitin ligase) create a positive-feedback signaling loop which decorates mitochondria with phosphor-ubiquitin chains. Activation and translocation of Parkin from the cytosol to the mitochondria involves its N-terminal phosphorylation and binding to Ser65-phophorylated ubiquitin [100]. The phosphorylation of both Parkin and ubiquitin is conducted by PINK1, after its stabilization on the outer mitochondrial membrane of dysfunctional/depolarized/damaged mitochondria. Mitochondria that are covered in ubiquitin are recognized by multiple autophagy receptors, which assemble the autophagosomal machinery for degradation of damaged mitochondria [101].

Regulation and progression of mitochondrial biogenesis (Figure 1) is coordinated by the master regulator peroxisome proliferator-activated receptor gamma coactivator 1-alpha (PGC1α). Sirtuin 1 (SIRT1) deacetylation or AMP-activated protein kinase (AMPK) phosphorylation of PGC1α initiates mitochondrial biogenesis [102,103]. Activated PGC1α stimulates nuclear factor erythroid 2-like 1 (NFE2L1 or NRF1) and nuclear factor erythroid 2-like 2 (NFE2L2 or NRF2) and increases the expression of the mitochondrial transcription factor A (TFAM) [104,105] which is the end-effector for the transcription and translation of mtDNA. Increasing the replication of mtDNA, the expression of mitochondrial proteins and translocation of mitochondrial proteins characterizes mitochondrial biogenesis.

**Figure 1 ijms-24-15788-f001:**
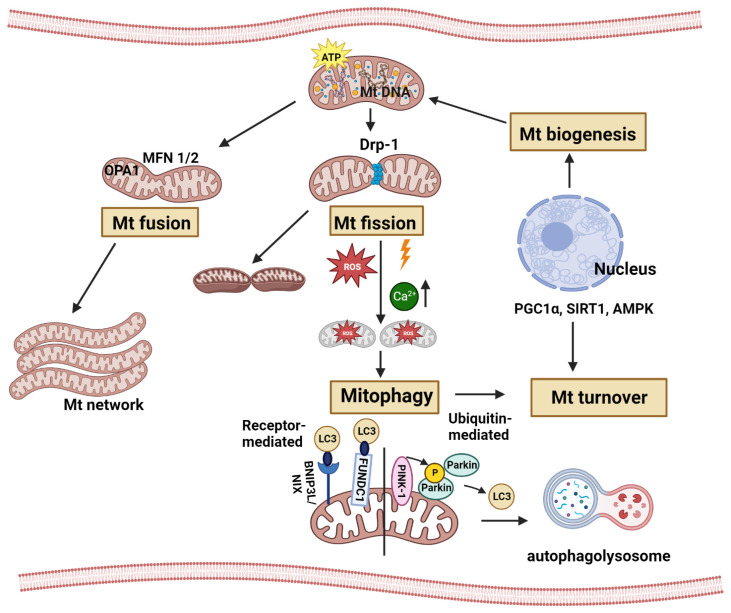
MQC pathways and main regulatory molecules essential for mitochondrial function, maintenance and cell homeostasis (created with BioRender.com). Mt undergo fission and fusion, oscillating between globular and elongated morphology. Mitochondrial fission is essential for the proper distribution of Mt into the daughter cells during division and serves mitophagy by generating “edible” smaller units from injured Mt. Excessive fission is both a major sign and an inducing factor of cell stress, and is associated with deterioration of ATP production, extensive ROS generation, apoptosis, or necrotic death. The key component of mitochondrial fission is Drp1 which, upon induction of fragmentation, can translocate from the cytosol to the outer mitochondrial membrane (OMM). The opposite process, fusion, is catalyzed by the Mfn1 and Mfn2 in the OMM and OPA1 in the inner mitochondrial membrane (IMM). The maintenance of mitochondrial network integrity is orchestrated by the processes of mitophagy and mitochondrial biogenesis. Regulation of Mt biogenesis is coordinated by PGC1α, SIRT1 and AMPK, increasing the replication of MtDNA and the expression of mitochondrial proteins. The key mitophagy pathways are divided into receptor-mediated mitophagy and ubiquitin-mediated mitophagy. During receptor-mediated mitophagy, under mitochondrial stress (i.e., hypoxia or depolarization), mitophagy receptors localize to the OMM. Both major mitophagy receptors, FUNDC1 and BNIP3L/NIX, facilitate phagophore formation. Ubiquitin-dependent mitophagy is PINK1/Parkin-dependent, and phosphorylation of both Parkin and ubiquitin is conducted by PINK1. Under Mt stress, PINK1 and Parkin decorate Mt with phosphor-ubiquitin chains. Mt covered in ubiquitin are recognized by multiple autophagy receptors that assemble the autophagosomal machinery for degradation of damaged Mt. LC3 is involved in both ubiquitin-dependent and receptor-mediated mitophagy.

### 2.3. Intercellular Mitochondrial Transfer Role in Tissue Homeostasis

It is well-established that Mt are dynamic organelles that undergo frequent changes in response to cellular demands [106]. In addition, it is now recognized that they can also move between cells, a process that contributes to rescuing cells with damaged Mt [107]. Intact Mt, or their components, could moderate intercellular communication between cells located in the same or different layers in the tissue (horizontal or vertical transfer, respectively), or even act on very distant cells [108]. This transfer of Mt and their components between mammalian cells can induce changes in mitochondrial genetic and functional profiles within the recipient cells [108,109]. Intercellular mitochondrial transport occurs through several mechanisms, including tunneling nanotubes (TNTs), gap junction channels, EVs, and cell fusion (please see the details in recent review papers) [110,111]. Specific molecular pathways and effectors in different types of mitochondrial transfer are summarized in the review by Guo et al. [112]. Future research will reveal how a particular mode of transport is chosen or favored. The determining factors likely include distance between cells, environmental factors, and the nature and severity of cell damage. It is also important to mention that cell-free mtDNA (cf-mtDNA), released by mitochondrial permeability transition pore opening, cell apoptosis, or death, may be recognized by cells as danger-associated molecular patterns (DAMPs), stimulating inflammatory signaling pathways [113]. However, it has to be noted that cf-mtDNA, as a part of intact Mt that circulate freely or as a part of EV cargo, does not have any inflammatory effect and is abundantly present in healthy subjects [113]. In tumor cells, Mt received from surrounding cells could have an important role in cancer survival, increased proliferation and invasion capabilities, contributing to metastasis and resistance to chemotherapy [112]. Therefore, the inhibition of Mt transfer could be used as a powerful target in cancer therapy. In addition, mitochondrial transfer modulation opens an avenue for promotion of normal cellular homeostasis and the restoration of a healthy energy balance and metabolism.

### 2.4. Mitochondrial Dysfunction

Mitochondrial defect/dysfunction are prominent in both genetic and acquired diseases, and mitochondrial transplantation, e.g., via MSCs, could be beneficial. In genetic conditions, mutations in mtDNA and/or nuclear DNA-encoding mitochondrial proteins can alter mitochondrial functions, leading to various inherited diseases [114]. Mitochondria are highly dynamic, as they undergo fission and fusion cycles, a process referred to as mitochondrial dynamics. These processes, as well as mitochondrial health, are in close correlation with cell metabolism, reactive oxygen species (ROS) production, and mechanical cellular alterations. As described above, mitochondrial health is maintained by the processes of MQC, aiming to optimize mitochondrial function. Since MQC, i.e., the regulation and maintenance of mitochondrial homeostasis, is essential not only for the fine-tuning of cell energy production [115] but also for overall cellular health, mitochondrial dysfunction and inadequate QC inevitably lead to metabolic disorders in many pathological conditions [116,117]. Defective MQC has been associated with diseases of the heart [118], lung [119,120], and kidneys [121], and in neuronal and muscular disorders [122]. Further, the metabolic switch from ATP production by oxidative phosphorylation to aerobic glycolysis (the Warburg effect) is one of the underlining mechanisms of cancer cell proliferation, allowing transformed cells to adjust and grow in a poorly oxygenated milieu [123].

The most-studied genetic mitochondrial disorder, caused by mtDNA m.3243A>G mutation, is responsible for complex genetic, molecular, and phenotypic pathological changes (see details in Lee et al.’s 2022 review paper [124]). Having only one functional copy of certain genes could also cause mitochondrial disease. Haploinsufficiency of AarF Domain Containing Kinase 2 (ADCK2) gene, which encodes the mitochondrial protein kinase family, generates coenzyme Q10 deficiency in skeletal muscle, leading to impaired physical performance [125]. Insufficient coenzyme Q10 production also increases oxidative stress, further potentiating mitochondrial dysfunction [126].

While genetic or acquired mitochondrial disorders differ in transmission and prevalence, in both cases mitochondrial transplantation could be considered as a therapeutic tool. For an optimal outcome, the effects of mitochondrial transplantation should be long-lasting in the genetic diseases, while short-term treatment may be sufficient in acquired disorders [127]. Mitochondrial transfer into damaged tissue could be achieved by direct transplantation of isolated healthy Mt, or by delivery using microbubbles, microvesicles, or cells, including MSCs. In fact, MSCs are an abundant source and powerful delivery system of healthy Mt that could be transferred to the injured/dysfunctional cells. Therefore, MSCs represent promising therapeutic options, especially for acquired mitochondrial dysfunctions, in both acute and chronic organ disorders. Mt transfer can ameliorate tissue injury and prevent extensive organ damage by improving mitochondrial function and MQC in recipient cells, thereby promoting repair [128,129]. Ischemia reperfusion injury (IRI) is an important acquired disorder where Mt transplant therapy in preclinical models has already been proven successful. This pathological process is common to a range of diseases and is a key event after trauma, transplantation, or major surgery. However, despite promising early studies, a better understanding of the mechanisms involved in Mt-transfer treatment is essential for translation into clinical settings [130].

## 3. Involvement of Mitochondria in MSC Therapeutic Effects

### 3.1. MSCs and Mitochondrial Dynamics

Optimal mitochondrial functions are crucial for MSC actions through a couple of key effects. First, optimal energy production by mitochondria is needed for all aspects of MSC biology. Second, the state of Mt is a crucial determinant of stemness. Third, the beneficial effects of MSCs on cells in the damaged organs depends on enhancing MQC, at least in part via mitochondrial transfer. This process also relies on optimal MQC and Mt dynamics in MSCs.

Regarding mitochondrial dynamics in stem cells, we emphasize three major points. (1) Rapidly proliferating cells, including various stem cells and cancer cells, prefer anaerobic energy production. Accordingly, they predominantly exhibit fragmented, small (low-OXPHOS) mitochondria [52,53,131,132,133]. Mitochondrial fragmentation also helps efficient transfer of mitochondria to the daughter cells during cell division, with the exception of neuronal stem cells with elongated Mt [134]. MSCs represent an intermediate phenotype: they contain a higher percentage of globular/spherical mitochondria than many somatic cells, but also more tubular mitochondria than pluripotent embryonic or epiblastic stem cells [51,53]. Although MSCs primarily utilize glycolysis, they are capable of OXPHOS, indicating a versatile metabolism [53,54]. (2) Differentiation is usually accompanied by a shift from spherical to tubular mitochondria. Moreover, mitochondrial dynamics seems to have a causal role in the process, as the inhibition of Drp1 was shown to abolish pluripotency and promote differentiation [135]. Conversely, Mfns were found to be necessary for differentiation, while their absence hindered the process [136]. However, this relationship is highly cell type- and context-dependent; for example, Drp1 activity is necessary for myogenic and chondrogenic differentiation [136,137]. Thus, it is more accurate to state that properly regulated mitochondrial dynamics is crucial for the various phases of the differentiation process [51,53]. Regarding MSCs, their osteogenic and adipogenic differentiation is accompanied with (and primed by) decreased fission and increased fusion, resulting in net elongation, whereas their chondrogenic commitment is characterized by the opposite processes and thus net fragmentation [136,138]. If the beneficial effects of MSCs in tissue regeneration are linked to their undifferentiated state, then their intermediate mitochondrial connectedness represents the optimal dynamics for this capacity. (3) Mitochondrial transfer from MSCs to the target cells is mediated by TNTs, EVs, and by gap junctions [139,140]. While the role of mitochondrial dynamics in mitochondrial transfer has not been extensively characterized, one can surmise that it is a significant factor. Namely, a large, interconnected network may be less amenable for efficient TNT-mediated transfer or vesicular packaging; on the other hand, the donor mitochondria have to be functionally intact and fusion-competent in order to support OXPHOS in the recipient cells [42,141,142,143]. Thus, it seems likely that optimal transfer requires an optimal size range and thus dynamics—an intriguing hypothesis for future research. Finally, mitochondria within MSCs are also exposed to environmental stresses that may lead to apoptosis or senescence. The presence of these changes can determine the suitability of MSCs to restore function via mitochondrial transfer. In summary, future studies should elucidate the complex roles of mitochondrial dynamics in MSC-derived mitochondrial transfer and QC, both in the donor and in the recipient cells.

### 3.2. Mitochondrial Transplantation as a Promising Therapeutic Tool

There is ample evidence highlighting the beneficial effects of MSCs on tissue injury, and many studies show that these effects require mitochondrial transfer. Based on these, transferring healthy Mt into injured tissue represents an attractive new therapeutic approach [144]. Mitochondrial transfer not only holds the potential to correct mitochondrial dysfunction, but also to activate metabolic or immunomodulatory pathways involved in the restoration of normal tissue function [145]. Mitochondrial transplantation could be achieved by the delivery of isolated Mt or by MSCs/MSC-derived EV injection. Further, Mt transfer is a bidirectional process that has an impact on both donor and recipient cells [144]. There are currently several clinical studies assessing the effects of Mt transplantation in genetic mitochondrial diseases, cerebral and myocardial infarction, graft rejection, and dermatomyositis [146]. One study (NCT05669144) is actually comparing the effects of isolated mitochondria and MSC-derived EVs in coronary diseases and exploring if co-transplantation of both could assure additional therapeutic benefits.

### 3.3. Mitochondrial Transfer as a Key Mechanism by Which MSC Achieve Their Therapeutic Benefits

The interaction of MSCs with their microenvironment is bidirectional. This is also true for mitochondrial exchange, one of the crucial mechanisms involved in MSC beneficial action. Namely, Mt from damaged cells are taken up and degraded by MSCs stimulating mitochondrial biogenesis in these cells. These effects increase the capacity of MSCs to donate their own Mt to injured cells. Dynamic transfer of Mt and/or mitochondrial DNA (mtDNA) between MSCs and damaged cells has been suggested by the early work of Spees et al. [107] as a potential therapeutic strategy. Transferred mitochondria considerably improve the function of acceptor cells [147], restoring their Mt content and homeostasis [25], while mtDNA captured by recipient cells is involved in cell repair [107]. In addition, MSCs also improve age-related phenotypes in cells of aging mice by changing growth factor and cytokine secretion and by increasing the number of Mt and the telomere length [148].

In inflammatory conditions and within the damaged tissue, MSCs respond to damage-associated molecular patterns (DAMPs) such as S100, heat-shock proteins, high-mobility group box 1 (HMGB1), ROS, or damaged Mt [143]. In such an environment, MSCs secrete anti-inflammatory molecules [149] and increase expression of reparatory, cytoprotective enzymes such as heme oxygenase-1 (HO-1) [5]. In addition, Mt from injured cells are engulfed and degraded by MSCs. This process is boosting Mt biogenesis in MSCs and enhances their capacity to donate Mt to injured cells, to regulate MQC, and to improve Mt function in recipient cells. This cascade of events has been demonstrated in MSC co-culture with distressed somatic cells such as cardiomyocytes or endothelial cells, but also in a model of myocardial infarction in vivo [150,151].

Thus, transfer of Mt is a central event in the effects of MSCs (Figure 2). The exact mechanisms and signaling pathways involved in Mt transfer from MSC/EVs to neighboring damaged and stressed cells are under intense investigation [28,31,152,153]. Proposed mechanisms include transfer via TNT, EVs, gap junctions, and cell fusion as well as other modes [31]. However, the underlying signaling mechanisms remain only partially explored. It has been demonstrated that activated stress signals in injured tissue trigger the transfer of Mt from MSCs or MSC-EVs to recipient cells, such are macrophages (Mϕ), epithelial, endothelial, and other cells. The steps and changes whereby Mt transfer restores optimal Mt functions in recipient cells also remain incompletely defined. Nevertheless, MQC regulation is certainly one of the most important mechanisms.

TNT formation between MSCs and damaged cells [154] largely depends on the mitochondrial outer membrane Rho-GTPases, Miro1 and Miro2 [155]. Interaction of these with other accessory proteins allows Mt movement on cytoplasmic tunnel-tube extensions that connect two cells [156,157]. These calcium-sensitive proteins bind Mt to the KLF 5 kinesin motor protein and other accessory proteins (like TRAK 1 and TRAK2), forming a motor-adaptor complex that contributes to Mt transfer. Indeed, Ahmad et al. [156] demonstrated that overexpression of Miro1 in MSCs augmented mitochondrial transfer and the rescue of epithelial injury, while Miro1 knockdown abrogated MSC efficacy. Zhang et al. [158] confirmed that the expression of Miro1 as well as MSC responsiveness to TNFα, are important for tunnel-tube formation and Mt transfer from MSCs to damaged cardiomyocytes.

In addition, reactive oxygen species (ROS) signaling has also been shown to be involved in tunnel-tube formation and Mt transfer between MSCs and corneal epithelial cells [159].

Another mode of Mt transfer to damaged cells is via MVs secreted by MSCs. For example, in the ARDS environment, MSCs ameliorated lung injury through EV-mediated mitochondrial transfer [160]. Accordingly, MSC-EVs are proposed as nano-therapeutic agents that could rescue mitochondrial damage in injured liver, kidney, spleen, and lung tissue [161]. Although MSCs could cross the blood–brain barrier (BBB) of an injured brain, MSC-EVs represent a more efficient cell-free treatment approach [162].

Nevertheless, despite these advances in our understanding of the role and mechanisms of Mt transplants, large gaps still remain in our understanding of this crucial effect.

## 4. MSC Mitochondrial Transfer in Organ Injury

### 4.1. Cardiac Injury and Effects of MSC Mitochondrial Transfer

In light of the prevalence of cardiovascular disease, there is a huge need to develop better therapies to aid repair of cardiac injury. MSCs have been proposed to augment this process, and data obtained in animal models of cardiac disease suggest that Mt transfer plays a key role in the beneficial effects in heart tissue (Figure 2). Specifically, in mice suffering from anthracycline-induced cardiomyopathy, MSCs that were able to induce more mitochondrial retention in cardiomyocytes generated better bioenergetic preservation and enhanced cardiac recovery (Table 1). Moreover, the outcome was dependent on TNT formation and Miro1 expression in MSCs [158]. In a cell culture model using human cardiomyocytes, doxorubicin-induced injury was ameliorated via MSC-EV treatment, an effect dependent on functional Mt in EVs [163]. Further, cardiomyocytes exposed to IRI were rescued in co-culture with MSCs [164], while direct intramyocardial transplantation of MSC-derived Mt (MSC-Mt) into the area of myocardial infarction (MI) improved heart function in mice. Interestingly, MSC-Mt were more effective in preserving cardiac function after MI then Mt derived from skin fibroblasts, suggesting unique properties of MSC-derived Mt [165]. In a hibernating myocardium model in juvenile swine, an MSC epicardial patch was able to mitigate the persistent decrease of myocardial function by improving Mt morphology and function in cardiac tissue [166].

In a murine model of ischemic heart disease (Table 1), MSCs that were preconditioned with Mt isolated from fetal cardiomyocytes showed increased therapeutic benefits [167]. This observation offers an alternative strategy for improving MSC efficacy by simple pre-incubation with Mt before their use for the treatment. Also, a demonstration that cardimyocyte Mt transfer to MSCs augmented their therapeutic potential for cardiac injury indicates that selection of Mt from the type of cells that MSCs will encounter in the environment may be important, further improving MSC efficiency.

Clinical trials with MSC or MSC-derived EVs are ongoing in patients with ischemic or toxic cardiomyopathy, fibrotic heart disease, or MI-induced heart failure. MSC or MSC-derived EVs have been proven to be safe and effective in many clinical trials for cardiac diseases (as detailed in Bhawnani et al., 2021) [168], but a lack of knowledge of optimal timing, dose, and route of delivery are still limiting the routine use of MSCs in clinical practice. MSC pre-conditioning, or genetic modification, or the use of MSC-EVs, may largely enhance the overall therapeutic benefits [169]. Analysis of samples taken from patients involved in clinical trials would allow a direct insight into the mechanisms of action of MSCs. Such an evaluation has already been performed in induced cardiomyocytes, generated from the SENECA phase 1 trial (NCT02509156) patients, injured with doxorubicin and treated with MSC-EVs. This study showed that only Mt-rich EVs with preserved Mt function were able to improve contractility, energy production, and mitochondrial biogenesis in damaged cardiomyocites [163], providing evidence of the importance of Mt tranfer in humans.

### 4.2. Lung Injury and Implication of MSC Mitochondrial Transfer

Acute and chronic lung disease, including ARDS, asthma, and pulmonary fibrosis, are promising areas for potential use of MSC therapy, and the role of Mt transfer is emerging in this context too (Figure 2). Bibliometric analysis of the most recent research regarding pathobiology, prevention, and therapeutic options in acute respiratory distress syndrome (ARDS) showed that the most frequent key words used in these publications are MSCs, Mt, mtDNA, mitophagy, and apoptosis [170].

Treatment of ARDS mice with alveolar Mϕs cultured with MSC-EVs ameliorated lung inflammation and injury. In the ARDS environment, MSCs interact with Mϕs, promoting an anti-inflammatory and highly phagocytic Mϕ phenotype through EV-mediated mitochondrial transfer. Mitochondrial transfer from MSC to Mϕs was demonstrated in a direct MSC/Mϕ co-culture, and this occurred, at least in part, through tunneling nanotube (TNT)-like structures (Table 1). Blockage of TNT formation resulted in failure of MSC-induced improvement of Mϕ bioenergetics and completely abrogated the effect on Mϕ phagocytosis in vitro [171]. TNT blockade was also shown to diminish the antimicrobial effect of MSCs in an in vivo E.coli ARDS mouse model [147]. Therapeutic effects of Mϕs co-cultured with MSC-EVs were prevented by rendering Mt in Mϕs or MSC-EVs dysfunctional using rhodamine 6G pretreatment [160]. In a similar ARDS mouse model, MSC-EVs reduced the extent of lung injury and enhanced alveolar– capillary barrier integrity, with a key role for mitochondrial transfer from MSC-EVs to injured cells, since EVs with dysfunctional Mt failed to achieve effects [172].

MSCs protected rat lungs against acute IRI (Table 1) by reducing inflammation, oxidative stress, mitochondrial damage, Drp-1 expression, cell apoptosis, and autophagy pathways [173].

MSCs also have beneficial effects in chronic pulmonary diseases, and their effect mainly depended on their ability to communicate with injured lung cells via secreted factors, EVs, and through the donation and transfer of Mt [174].

iPSC-derived MSCs ameliorated mitochondrial dysfunction caused by cigarette smoke-induced oxidative stress in human airway smooth muscle cells (ASMCs) and in a mouse model of chronic obstructive pulmonary disease (COPD), thereby reducing airway inflammation and hyperresponsiveness (Table 1). These effects were mediated by mitochondrial transfer and by paracrine factors secreted by MSCs [175]. In a similar COPD model (Table 1), where the efficacy of MSCs, MSC-EVs, or combination of these, was tested, the authors showed that cigarette smoke-induced lung damage was associated with changes in genes involved in mitochondrial fission/fusion, along with altered levels of cytokines. The MSC + MSC-EVs combination treatment showed additional protective effects compared to MSC or MSC-EV treatment alone. The combination treatment increased expression of the mitochondrial fusion proteins (mitofusin 1 & 2), led to mitochondrial-transfer enhancement, and decrease of inflammation, thus reducing bronchial epithelial damage [176].

Mitochondrial dysfunction is also present in and may be an important factor in the pathogenesis of asthma (Table 1). Accordingly, MSCs were able to ameliorate inflammation and hyperresponsivity of bronchial epithelial cells via mitochondrial transfer to damaged cells, and this process was dependent on Miro1, a mitochondrial Rho-GTPase [156]. This mechanism was proven in vivo as MSCs overexpressing Miro1 rescued airway epithelial injury in an asthma mouse model to a greater extent than control MSCs, while Miro1 knockdown led to a loss in MSC efficacy [156]. Similarly, MSC therapy was able to alleviate inflammation and improve Mt function in asthmatic mouse lungs, thereby preventing lung damage and asthma immunopathology [177]. MSCs were also shown to react to exposure of the serum of asthma patients. When pre-conditioned with serum from asthma patients, MSCs released more anti-inflammatory factors, had reduced intrinsic mitochondrial respiratory capacity, and induced a greater Mϕ shift toward an M2-like phenotype, compared with MSCs cultured without the serum. Further, in a mouse allergic asthma model, these pre-conditioned MSCs were more effective than naïve MSCs, leading to further reduction of both lung inflammation and remodeling [178].

MSC-EVs from healthy (H) but not emphysematous donors (E) were able to reverse cardiopulmonary dysfunction in mice with severe emphysema. Mt from E-EVs were more elongated, less functional, and produced more ROS compared with Mt from H-EVs. Notably, in this model (Table 1), treatment with MSC-EVs offered an advantage over MSCs, causing better lung-function recovery [179].

Bronchopulmonary dysplasia (BPD) is a chronic lung disease affecting premature newborns, who need oxygen therapy. In vitro testing of endogenous MSCs obtained from the umbilical cord of extremely low body-weight (ELBW) infants has revealed they have mitochondrial dysfunction and dysregulated MQC, low ATP production, and decreased mitochondrial survival (Table 1). These characteristics were associated with a bronchopulmonary dysplasia (BPD) risk in ELBW infants. Since these newborns receive oxygen therapy, it is notable that hyperoxia-exposed MSCs from infants who died or developed BPD produced more ROS, had lower PINK1 expression, and deficient mitophagy. The authors speculated that such mitochondrial abnormalities may eventually lead to a depletion of the endogenous MSC pool and cause disruption of lung development in ELBW infants suffering from BPD [180].

Mitochondrial dysfunction, along with profibrotic factor release and signs of senescence, have also been demonstrated in endogenous BM-MSCs isolated from patients with idiopathic pulmonary fibrosis (IPF) [181].

Both phase I and II clinical trials using MSCs or MSC-EVs were completed or are underway to test effects in acute [182] and chronic lung diseases [174]. These trials are showing promising results. It appears that mitochondrial transfer to injured cells and Mϕs by MSCs/MSC-EVs is one of the most important mechanisms of beneficial effects in pulmonary disorders. However, further exploration of these and other mechanisms in different stages of the disease is needed. Moreover, a better understanding of the effects of the lung environment composition within a particular patient is crucial to further improve MSC therapy. This knowledge should lead to the design of case-specific regimens and improved protocols of MSC/MSC-EV production. Ultimately, these are necessary steps for adding MSC therapy to the clinical routine.

### 4.3. Liver Injury and Modes of MSC Mitochondrial Transfer

Promising preclinical studies (Table 1) and clinical trial results also raised hopes for a beneficial MSC therapy in liver diseases (Figure 2). MSCs may prove beneficial in ischemic liver damage and chronic liver diseases like non-alcoholic fatty liver disease (NAFLD) and its advanced form, steatohepatitis.

There is a strong need for new therapies for acute liver damage. Hepatic diseases and especially liver IRI involve dynamic processes that induce liver dysfunction and contribute to mortality during liver transplantation, major liver resection, and hemorrhagic shock [183]. Mitochondrial dysfunction plays a central role in liver IRI [33]. The mitochondrial network was shown to be crucial for energy distribution and also for communication between MSCs and injured liver cells [31]. Promising data show that MSCs ameliorated liver damage and hepatocellular apoptosis in mouse liver IRI [7]. In these studies, a role of the MSC capability to regulate mitochondrial quality control through Parkin/PINK1 dependent mitophagy was verified. An MSC-conditioned medium (CM containing secreted EVs, Mt, and cytokines) used in in vitro experiments also activated the AMPKα pathway and restored Mt energy production, suppressed hepatocellular apoptosis, and inhibited Mt ROS accumulation. These beneficial effects of MSC-CM were partly blocked by silencing PINK1 or by using the AMPK inhibitor dorsomorphin [7]. MSC EVs were also shown to offer beneficial effects in multiple studies involving liver conditions (Table 1). In a rodent liver, IRI MSC-derived EVs ameliorated damage through Mt transfer to hepatocytes, reducing oxidative stress and neutrophil infiltration in the liver [184]. Different studies also suggest that the effects may have been at least in part due to modulating neutrophil functions. MSC-EVs improved morphology and function of hepatocytes and decreased NET formation in peri-operative IRI in human liver grafts. In a mouse IRI model, liver inflammation was ameliorated by inhibiting NETosis. This was achieved through Mt transfer from EVs to neutrophils, which triggered Mt fusion and restored Mt function in neutrophils [185].

Strong evidence that MSCs can improve liver mitochondrial functions in vivo came from a study using rats with D-galactose induced hepatic mitochondrial dysfunction (Table 1). In this model, umbilical cord (UC)-MSCs enhanced Mt respiration, energy production, and boosted antioxidant capacity via activation of the Nrf2/HO-1 pathway [186].

MSCs might also improve mitochondrial functions in chronic liver disease (Table 1). Liver cirrhosis in rats induced by bile duct ligation (BDL) was alleviated by placenta-derived MSCs, particularly when phosphatase of regenerating liver-1 (PRL-1) expression was enhanced in MSCs (MSC-PRL-1). Compared with naïve cells, MSC-PRL-1 led to a stronger improvement of the mitochondrial metabolic state, including ATP production and mitochondrial biogenesis. These MSCs had better engraftment in injured liver, induced a further increase of MtDNA and ATP in hepatocytes, and had an enhanced anti-fibrotic effect compared to naïve cells [187]. MSC-PRL-1 cells were also more efficient than naïve MSCs in improving oxidative stress and mitochondrial metabolism in the same rat liver cirrhosis model [188].

Nonalcoholic fatty liver disease (NAFLD) and its advanced form, steatohepatitis, represents the most common global liver illness, with serious risk factors for cirrhosis, hepatocellular carcinoma and diabetes [189]. In an NAFLD and diabetes mouse model, MSC-CM improved insulin resistance, preserved liver structure, and reduced inflammation, while enhancing antioxidant capacity and Mt function. MSC-CM treatment of hepatocytes in a cell culture subjected to lipid accumulation, decreased cell apoptosis and improved mitochondrial energy production. The effects were mediated by Sirtuin 1 (Sirt1), a positive regulator of liver X (LXR) receptors that orchestrate cholesterol and lipid homeostasis [190]. In a similar mouse model of steatohepatitis, human MSCs resolved lipid load by donating Mt to hepatocytes [191,192], thereby increasing mitochondrial abundance and oxygen consumption, thus improving metabolic liver function and reducing inflammation [193].

MSC therapy also holds promise for the treatment of inherited liver diseases. Namely, MSCs co-cultured with skin fibroblasts isolated from inherited mitochondrial disease patients, rescued impaired mitochondrial morphology and function, shifting the fission state of Mt to their more fused appearance [193].

MSC/MSC-EVs are showing promising results in completed or ongoing hepatic disease clinical trials, especially in chronic liver failure, NAFLD or alcoholic hepatitis, liver cirrhosis, and hepatocellular carcinoma [194,195]. In a randomized controlled trial in decompensated liver cirrhosis, MSC treatment was found to be safe and effective, and markedly improved liver function during 48 weeks of follow-up [196]. Similarly, in a small phase I–II trial, MSCs improved liver function in patients with acutely decompensated liver cirrhosis [197]. MSCs were also successfully used as a preventive treatment for acute liver allograft rejection in some pilot studies [198,199]. However, large-scale randomized trials with mechanistic studies conducted on patient samples are warranted. In these future studies, special attention should be given to Mt transfer from MSC/MSC-EVs to donor cells and hepatocyte Mt functions. While an improved Mt state and mitochondrial QC are crucial for each organ function, they are especially important for the liver’s metabolic and regulatory role.

**Table 1 ijms-24-15788-t001:** Preclinical studies describing implication of MSCs and MSC-EVs on mitochondrial transfer and MQC in cardiac, pulmonary, and hepatic injury.

Organ	Preclinical Model	Treatment	Outcome	Mechanism	References
**Heart**	Anthracycline-induced cardiomyopathy in mice	iPSC-MSCs and BM-MSCs	iPSC-MSC preserved heart tissue better than BM-MSCs.	Efficiency of Mt transfer TNT formation MSC expression of Miro1	Zhang et al., 2016 [158]
	Doxorubicin-induced injury in human cardiomyocytes in vitro	Co-culture or direct contact with MSC-EVs	Large MSC-EVs (>200 nm) ameliorated cardiomyocyte injury. Inhibition of Mt function in MSC-EVs attenuated efficacy.	Improved contractility ↑ ATP production ↑ Mt biogenesis	O’Brien et al., 2021 [163]
	Cardiomyocytes (H9c2) IRI in vitro	MSC-H9c2 co-culture	Marked resistance against IRI	↓ apoptosis ↑ Mt transfer from MSC to H9c2 via TNT formation	Han et al., 2016 [164]
	MI in mice	MSC-Mt transplanted in peri-infarct area	MSC-Mt preserved better cardiac function after MI then fibroblast derived Mt.	↑ Vessel density in MI area ↓ Apoptosis and endothelial cell senescence (via ERK pathway activation) ↓ Heart remodeling	Liang et al., 2023 [165]
	Co-culture of MSCs and cardiomyocytes or endothelial cells pretreated with H_2_O_2_. MI in mice	MSCs	Increased capacity of injured cells to combat oxidative stress. Reduced damage of infarcted mouse hearts.	Mitochondrial exchange between MSCs and damaged cells ↑ HO-1 ↑ mitochondrial biogenesis	Mahrouf-Yorgov et al., 2017 [150]
	Hibernating myocardium model without infarction in juvenile swine (surgical stenosis of the left anterior desc. coronary artery)	Epicardial MSC patch applied during coronary artery bypass graft	Improved myocardial function (measured by cardiac magnetic resonance imaging). Improved Mt function.	Improved Mt morphology ↑ Mt biogenesis and ATP production in cardiac tissue	Hocum et al., 2021 [166]
	MI in mice	MSCs preconditioned or not with Mt	MSCs primed with Mt had increased capacity to repair mouse myocardial infarct.	↑ Mitophagy of exogenous Mt ↑ Anti-inflammatory, proangiogenic and anti-fibrotic properties of MSCs primed with Mt.	Vignais et al., 2023 [167]
**Lung**	MSC/alveolar Mϕ direct co-culture	BM-MSCs	↑ Mϕ oxidative phosphorylation and phagocytosis	Mt transfer from MSCs to Mϕ by TNT formation	Jackson et al., 2017 [171]
	*E. coli* mouse ARDS model	BM-MSCs	↑ Mϕ phagocytosis Antibacterial effect	Mt from MSC aquired by lung Mϕ through TNT formation ↑ Phagocytic activity of Mϕ positive for MSC Mt	Jackson et al., 2016 [147]
	Mϕ and MSC non-contact co-culture stimulated with LPS or BALF from ARDS patients ARDS mouse model	MSCs Alveolar Mϕ cultured with and without MSC-EVs	↓ Cytokine production ↑ M2-like Mϕ marker expression ↓ Pulmonary inflammation ↓ Lung injury	↑ Mϕ phagocytic capacity Involvement of Mt in Mϕ & CD44 in MSC-EVs Crucial role of Mt in Mϕs and MSC-EVs	Morrison et al., 2017 [160]
	Primary cells and human lung cuts exposed to endotoxin or plasma of ARDS patients (in vitro) ARDS mouse LPS model	MSC-EVs	Improvement of increased cell permeability and Mt dysfunction ↓ Lung injury Restoration of alveolar-capillary barrier	Normalization of oxidative phosphorylation EVs with dysfunctional Mt was ineffective. Mt transfer and restoration of Mt function	Dutra–Silva et al., 2021 [172]
	Acute lung IRI rat model Lung epithelial cell line exposed to H/R injury (in vitro)	AD-MSCs and iPSC-MSCs	Similar lung protection with both AD-MSCs and iPSC-MSCs ↓ Lung injury score ↓ Inflammation cells	↓ Mitochondrial damage/cell apoptosis, autophagy, and oxidative stress ↓ Drp-1, Mt Bax/caspase3/8/9 and authophagy pathways (in vitro)	Lin et al., 2020 [173]
	ASMCs exposed to cigarette smoke media COPD mouse model (exposure to ozon)	iPSC-MSCs	↓ Mt ROS ↓ Airway inflammation and hyperresponsiveness	Mt transfer to donor cells ↑ Mt function	Li et al., 2018 [175]
	COPD mouse model (mice exposed to cigarette smoke for 10 days) BEAS2B-mMSC co-cultures	MSCs, MSC-EVs, MSC + MSC-EVs	↓ Bronchial epithelial damage ↓ Inflammatory cellular infiltration	↑ Mitofusin 1 and 2 ↑ Mt transfer ↓ Pro-inflammatory cytokines Same changes confirmed in co-culture settings	Maremanda et al., 2019 [176]
	Asthma mouse model	MSCs (naïve, over-expressing or knockout for Miro-1)	↓ Allergic inflammation and hyperresponsiveness of airways ↓ Lung injury	Mt transfer from MSCs to bronchial epithelial cells; Rho-GTPase Miro1-dependent process	Ahmad et al., 2014 [156]
	Asthma mouse model	BM-MSCs	↓ Lung inflammation ↓ Goblet cells mucus hyper-production Improved lung morphology	↓ Eosinophils and allergo-inflammatory cytokines ↓ Asthma induced mitochondrial gene expression ↑ Mt function	Huang et al., 2021 [177]
	Asthma mouse model	MSCs preconditioned or not with serum of asthma patients	↓ Lung inflammation ↓ Lung fibrosis ↑ Lung tissue regeneration	↑ Expression of TGFβ, IDO-1, TSG-6 by hMSC-serum ↑ fission ↓ respiratory capacity of Mt ↑ MSC apoptosis and their phagocytosis by Mϕ ↑ M2 Mϕ polarization	Abreu at al., 2023 [178]
	Co-culture of alveolar Mϕ and MSCs or MSC-EVs (in vitro) Mouse severe emphysema model	MSCs or MSC-EVs from healthy (H) and emphysematous (E) donor mice	Immunomodulatory effects ↑ IL-10 Improvement of cardiorespiratory dysfunction with MSC-EVs only from H donors	Abnormal Mt in E-MSCs and E-EVs—elongated, less functional and produced ↑ ROS vs. Mt from H-MSCs and H-EVs H-EVs showed better efficacy in comparison with H-MSCs, since they could access smaller airways, unreachable for MSCs.	Antunes et al., 2021 [179]
	ELBW infants with/without BPD (n = 39) UC-MSCs taken and studied in vitro	Endogenous MSCs isolated	MSCs with: Mt dysfuction ↓ ATP production and mytophagy ↓ Mt survival associated with BPD risk in ELBW infants.	Mt abnormalities may cause endogenous MSC pool depletion and disruptions in ELBW infant lungs.	Hazra et al., 2022 [180]
	IPF patients IPF (bleomycin) mouse model	Endogenous BM-MSCs isolated from IPF patients and age-mathed controls	BM-MSCs from IPF patients have signs of senescence with Mt dysfunction and DNA damage IPF BM-MSCs secreated pro-fibrotic factors and increased illness severity and inflammation in mice	Mt fragmentation ↓ Mt oxygen consumption and bioenergy ↑ IL-6, IL-8, IL-1β ↑ Pro-fibrotic factors	Cardenes et al., 2018 [181]
**Liver**	IRI in mice Hepatocites in vitro subjected to H/R injury	MSCs MSC-CM	↓ Liver injury Improved liver function ↓ Hepatocellular apoptosis	↓ ROS in tissue ↑ Parkin/PINK1 mitophagy ↑ ATP production by AMPKα activation	Zheng et al., 2020 [7]
	IRI in rats Hepatocyte cell culture	MSC-EVs	↑ Hepatic recovery ↓ Neutrophil infiltration and respiratory burst ↓ Oxidative stress	Mt transfer and Mt-located antioxidant enzyme (manganese superoxide dismutase (MnSOD) ↓ ROS—induced hepatocyte apoptosis and cell death in vitro	Yao et al., 2019 [184]
	IRI in specimen of human liver grafts (peri-operative) Mouse IRI	MSC-EVs	Liver graft morphology and function were better preserved Improved liver IRI in mice	↓ Liver graft inflammation and NET formation ↓ NET formation ↓ Netosis Mt transfer from EVs to intrahepatic neutrophils and their MQ control regulation	Lu et al., 2022 [185]
	D-Galactose induced hepatic disorder in rats Mt isolated from liver	UC-MSCs	Improved liver morphology and function Improved Mt respiration, ΔΨm and ATP production	↓ Histological lessions and liver enzymes ↑ Mt bioenergy and antioxidant capacity through Nrf2/HO-1 pathway	Yan et al., 2017 [186]
	Rat BDL cirrhosis	MSCs, naïve and tranducted with PRL-1	Liver regeneration Improved liver function Better efficacy of MSC-PRL-1 compared to naïve MSCs	↑ Metabolic state and Mt biogenesis of MSC-PRL-1 ↑ Engraftment, Mt DNA, biogenesis and ATP production in hepatocytes	Kim et al., 2020 [187]
	Rat BDL cirrhosis	MSCs, naïve and tranducted with PRL-1	↑ Oxidative capacity of MSC-PRL-1 MSC-PRL-1 vs. naïve MSCs improved further liver function and had greater antifibrotic effect	↑ Mt biogenesis and Mt lactate of MSC-PRL-1 MSC-PRL-1 have greater antioxidant effect vs. naïve MSCs	Kim et al., 2023 [188]
	Mouse NAFLD and diabetes (induced by high fat diet and streptozotocin) Hepatocytes treated with palmitic acid (in vitro)	MSC-CM	Improvement of insulin resistance and liver morphology ↓ Liver inflammation ↓ Cell apoptosis	↑ Liver antioxidant capacity ↑ Mt function in liver cells Crucial role of Sirt1 in cell protection	Yang et al., 2021 [190]
	Steatohepatitis in mice Hepatocytes and MSC co-culture	BM-MSCs	↓ Hepatocyte lipid content of ~40% ↓ Mt and peroxisomal dysfunction	Donation of Mt to hepatocytes Mt transfer from MSCs to hepatocytes by TNT formation	Hsu et al., 2020 [192]
	Steatohepatitis in mice	MSCs	Switch from liver lipid storage to its utilization	Donation of Mt to the hepatocytes Restoration of hepatocyte metabolism and oxidative capacity	Nickel et al., 2022 [191]
	Steatohepatitis in mice	MSCs	Improved liver morphology and metabolic function	Improvement of impaired Mt morphology and function ↑ Liver metabolic capacity and host gene shifting	Newell et al., 2018 [193]
	MSCs + fibroblasts (from Mt disease patients) −co-culture	MSCs	Improved Mt morphology and function	Inverting abundance of Mt fission toward fussion Mt state	Newell et al., 2018 [193]

Legend: ↑ increase; ↓ decrease.

### 4.4. Renal Injury and MSC Mitochondrial Transfer

Acute kidney injury (AKI) remains a disease with high mortality, while chronic kidney disease (CKD), which can be caused by highly prevalent conditions such as diabetes and hypertension, is an increasing public health concern. Therefore, the effects and use of MSCs have been explored in kidney-disease models. Whether MSCs can home to injured kidneys is still controversial, as some papers showed MSC homing in a renal IRI model [200,201], while others suggested the opposite in the same IRI model [202]. However, there is no doubt that MSCs can be beneficial in kidney disease and that mitochondrial transfer plays a key role in MSC reparatory effects in this organ (Figure 2). Preclinical animal models of AKI and CKD found that the therapeutic efficacy of both MSCs and MSC-EVs is mainly due to the cargo they deliver, consisting of Mt and biomolecules [203,204]. The treatment was shown to promote tubular epithelial cell recovery and angiogenesis, to induce mitophagy and Mt biogenesis, and to inhibit oxidative stress, inflammation, and fibrosis.

Several acute kidney injury models explored mechanisms of action (Table 2). MSC-EVs exhibit a protective role in both mouse renal IRI and tubular cell hypoxia/reoxygenation (H/R) models [205]. Mechanisms implicated in this protection include an exosome delivery that attenuated inflammasome (NLRP3) activation through miR-223-3p and consequent mitophagy enhancement by PINK1/PARKIN pathway induction. MSC-EVs also reduced Mt and mtDNA damage and inflammation after AKI, the effect partially dependent on the mitochondrial transcription factor A (TFAM) pathway [161].

Beneficial effects of MSCs on the tubules appear to be key for protection. UC-MSCs transplanted into mice with toxic cisplatin-induced AKI stimulated injured tubular cells to regain mitochondrial mass. PGC1α expression, NAD+, and Sirtuin 3 (SIRT3) activity were increased in these cells, enhancing antioxidant defense and mitochondrial ATP production [206]. Tracking of MSC-EVs in a renal IRI injury mouse model confirmed that most were taken up by proximal tubular epithelial cells [8]. The accumulated MSC-EVs increased MtDNA copy number, normalized mitochondrial membrane potential (ΔΨm), stimulated mitochondrial energy production, and antioxidant defense via the Keap1 (Kelch-like ECH-associated protein 1)—Nrf2 (the nuclear factor erythroid 2-related factor 2) signaling pathway.

Studies suggest a potential benefit from MSCs in CKD too (Table 2). In a rat CKD model, administration of MSCs decreased fibrosis and improved renal function. Cell-stress signaling pathways, activated in CKD animals, were down-regulated by MSCs therapy, thus decreasing stress/ischemic tissue damage [207]. While EVs derived from stem cells that contained Mt were involved in the therapeutic effects in CKD models [204], the exact role of Mt transfer and improved mitochondrial function remains to be established in the beneficial MSC effects in CKD.

There are several ongoing clinical studies using MSCs/MSC-EVs, mainly in diabetic kidney disease and chronic renal failure, although only a few are recruiting patients with AKI. Treatment efficacy, obstacles, and strategies will be elucidated, once the results of these studies become available.

### 4.5. Neurological Injury and MSC Mitochondrial Transfer Involvement

Neurological pathologies, such as those caused by traumatic brain injury (TBI) and neurodegenerative diseases, represent a serious health problem worldwide, affecting millions of people each year. These disorders are the foremost cause of disability and the second leading cause of death worldwide [208,209,210].

Neurological disorders were shown to be associated with two prominent pathological features, neuroinflammation and mitochondrial dysfunction. In fact, the connection between these was just recently recognized [211,212]. Neuronal Mt play a role in the regulation of astrocytic function and microglial activation, which are involved in immune responses and dead neuron clearance.

Brain damage, especially in ischemia-induced injuries (e.g., stroke), has a major effect on Mt. It prolongs mitochondrial permeability transition pore (mPTPs) opening, leading to excessive ROS release and perpetuation of mitochondrial dysfunction [213]. The consequent ATP deficiency and ROS surplus hamper brain-tissue recovery, ultimately causing irreversible cellular damage and death. MSC treatment [208] and the release of healthy Mt from these cells, or the transplantation of Mt isolated from MSCs into damaged brain tissue, are powerful tools that could ameliorate neurotrauma (Figure 2). Thus, correcting mitochondrial dysfunction and/or injecting Mt (isolated from healthy cells or released from injected MSCs or their EVs) in a neuronal injury may represent a promising novel therapeutic approach.

Cell culture models verified protection against neuronal cell death by MSCs (Table 2). Apoptosis was reduced and ΔΨm was restored in injured neuron-like PC12 cells co-cultured with MSCs [214]. In this model, mitochondrial transfer from the MSCs to PC12 cells occurred via tunneling nanotubes.

Data from a variety of cellular and animal models of TBI point to improved Mt functions following MSC therapy (Table 2). Injection of MSC-derived Mt in the brain was shown to be beneficial in TBI preclinical studies. Specifically, Mt therapy stimulates regeneration of injured neurons as demonstrated in a TBI in vitro model. The effects of injection of healthy Mt included recovery of the injured hippocampal neurons and restoration of their membrane potential [215]. Adipose MSCs (Ad-MSCs) or MSC-derived exosomes, injected intra-cerebroventricularly in rats with TBI, also facilitated functional recovery [216]. Another study using a rat TBI model and the injection of MSC-derived Mt in brain ventricles showed improved motor function via rescuing neuronal cells from apoptosis, promoting neuron formation, and reducing astrogliosis and microglial activation [217].

An interesting example for bidirectional Mt transfer was shown in a mouse model of transient focal cerebral ischemia. The study found that neurons could transfer impaired Mt to astrocytes for recycling. Functional Mt released from astrocytes were then taken up by adjacent neurons, amplifying cell-survival signals and promoting neurorecovery [218].

The function of microglia was also shown to be affected by MSC-derived factors (Table 2). In vivo and in vitro experiments revealed that exosomes taken up by microglia/Mϕ suppressed their activation by inhibiting the nuclear factor κB (NFκB) and p38 mitogen-activated protein (MAP) kinase signaling pathways. It also promoted the polarization of microglia towards the anti-inflammatory, pro-reparatory (M2-like) phenotype [216]. Notably, benefits of MSC treatment exist not only in the brain, but also in other parts of the nervous system. For example, in a spinal cord injury (SCI) rat model, Mt transfer from BM-MSCs to the injured spinal cord improved the locomotor function of injured rats [219].

One aspect of the beneficial effects appears to be protection against oxidative injury, which is an important pathogenic factor in brain injury of various origins. Mt transfer from MSC to mouse neurons in vitro increased neuronal survival and improved cell metabolism following oxidative damage (Table 2). Increasing expression of genes involved in TNT formation and Mt transfer (e.g., Miro1) in MSCs improved their benefits after neuronal oxidant injury [157]. MSC-EVs (containing Mt) showed therapeutic efficacy in oxidative damage of hippocampal neurons in vitro and in a seizure mouse model. The treatment restored the Nrf2 defense system and Mt dysfunction in an oxidative insult [220].

A key mechanism whereby MSCs provide neuroprotection (Figure 2) is the stimulation of autophagy and mitophagy in injured cells. Autophagy, including mitophagy, is a critically important process that facilitates neuronal recovery after TBI [221]. Accordingly, its malfunction is involved in neurodegenerative disease progression [222]. MSCs given to rats with TBI provided neuroprotection by inducing autophagy, mitophagy, and by elevating anti-inflammatory cytokines, such as interleukin-10 (IL-10) [223]. Therefore, transplantation of MSCs and MSCs transfected with IL-10 (MSC-IL-10) have protective effects against TBI pathology, with the latter exerting a stronger effect. Neuroprotection achieved through autophagy and mitophagy is mediated by the PI3K/Akt/mTOR pathway that induces efficient cell survival [223]. It also seems that IL-10 is an important player in mitophagy control in TBI, as a significant increase in mitophagy markers, such as NIX/BNIP3L (NIP3-like protein X, NIX), FUN14 domain containing 1 (FUNDC1), PTEN-induced kinase 1 (PINK-1), and hypoxia-inducible factor 1-alpha (HIF-1α), was noticed following transplantation of MSCs-IL-10 [223]. This notion is corroborated by a study showing that skeletal muscles of IL-10 knockout mice have higher levels of damaged mitochondria with disrupted ultrastructure compared to age matched wild-type controls [224]. Successful execution and completion of mitophagy is also beneficial in the context of cognitive functions’ preservation after TBI. Elimination of irreparably damaged Mt is activated as an early response to TBI and is essential for the reparative plasticity of injured brain cells [221]. Augmentation of this process therefore has strong benefits.

MSC and/or MSC-EVs treatment also holds promise for the treatment of neurodegenerative diseases [208], such as Alzheimer’s disease (AD), multiple sclerosis (MS), and Parkinson’s disease. Mt transfer to neuronal cells in AD models boosts ATP production, antioxidant release, and clearance of protein aggregates (Table 2). In an AD cellular model, UC-MSC conditioned media abundant in Mt alleviated mitochondrial oxidative stress and improved Mt function in damaged neuronal recipient cells [225]. In an AD mouse model, treatment with MSCs or their conditioned media ameliorated neuroinflammation-related cognitive impairment, including memory loss and improved Mt function in damaged brain cells [226]. MSC transplantation increased mitochondrial biogenesis in damaged neuronal cells, in a demyelination mouse model of MS too. In this model, MSC treatment increased the expression levels of transcripts involved in Mt biogenesis (PGC1α, NRF1, MFN2, and TFAM), improved mitochondrial homeostasis, and promoted antioxidant response [227].

The effects of MSCs or MSC-EVs (mostly exosomes) are currently being tested for patients with SCI, stroke [228,229], and neurodegenerative diseases [230,231] to evaluate their safety, tolerability, and treatment effects. Most of these studies are phase I (safety) trials with a small sample size. Some completed studies confirm that the treatment is safe and well-tolerated [230], and for these currently follow-up studies (phase IIa) are underway. Studies also explored different ways of delivery: MSCs or MSC-EVs have been injected intracerebroventricularly [230], intranasally [228], intravenously (i.v.), or intrathecally (ITh) [231].

Importantly, no serious adverse events in patients have been observed so far. However, large-scale controlled double blinded studies are needed to ascertain the therapeutic efficacy of MSCs for neurological pathologies. Bang et al. [229] conducted the first randomized trial in ischemic stroke, with patients injected with autologous MSCs (*n* = 39) or subjected to the standard treatment (*n* = 15). They showed that MSC treatment led to a five-fold increase in circulating EVs. Increase in EVs directly correlated with motor function improvement and neurogenesis/neuroplasticity indices determined by magnetic resonance imaging (MRI). In patients with progressive MS, autologous MSC transplantation was given via the i.v. or ITh route [231]. Treatment was well-tolerated and ITh cell administration induced additional benefits on the relapse rate, lesions, and cognitive changes. However, the study was designed to follow only the short-term effects of MSCs in the active disease state. Phase II and III trials are necessary to confirm these findings and to determine the long-term effects of MSC or MSC-EV administration in MS and patients suffering from other neurodiseases.

**Table 2 ijms-24-15788-t002:** Preclinical studies describing implication of MSCs and MSC-EVs mitochondrial transfer and MQC in renal and brain injury.

Organ	Preclinical Model	Treatment	Outcome	Mechanism	References
**Kidney**	Mouse renal IRI Tubular cell H/R	MSC-EVs	↓ Renal injury Renal function improvement	EV delivered miR-223-3p ↓ Inflammasome activation ↑ Mitophagy	Sun et al., 2022 [205]
	Mouse AKI Injured tubular cells	MSC-EVs	↓ Renal lesion formation ↓ Inflammation	↑ Mt energy production ↓ MtDNA damage and/or deletion TFAM pathway	Zhao et al., 2021 [161]
	Mouse AKI	UC-MSCs	↑ Renal repair ↓ Tubular damage	↑ Mt energy production ↑ Mt biogeneses ↑ PGC1α, NAD+, SIRT3	Perico et al., 2017 [206]
	Mouse renal IRI	MSC-EVs (tracked)	↓ Oxidative kidney damage ↑ Kidney function	↑ MtDNA copy number ↑ Mt ΔΨm stabilization and energy production Keap1-Nrf2 pathway	Cao et al., 2020 [8]
	Rat CKD	MSCs	Preserved residual renal function ↓ Kidney fibrosis	↓ Cell-stress signalling pathways (PI3K/Akt, pERK1/2, p-PKC and others)	Sheu et al., 2020 [207]
**Brain and** **Neurons**	MSC and injured PC-12 cells In vitro	Co-culture	↓ Apoptosis ΔΨm restoration	Mt transfer from MSCs to PC-12 by TNT formation	Yang et al. 2020 [214]
	Rat TBI and n vitro	Ad-MSCs and MSC-derived exosomes	Improved functional recovery	↓ Neuroinflammation ↑ Microglia/Mϕ shift to M2-like phenotype	Chen et al., 2020 [216]
	Rat TBI	Mt isolated from UC-MSCs	Improved motor function	↑ Neuronal formation and reparation ↓ Astrogliosis ↓ Microglial activation	Bamshad et al., 2023 [217]
	Rat contusion SCI	BM-MSCs	↑ Remyelination and functional recovery of motor neurons in SCI rats.	Mt transfer from BM-MSCs to injured motor neurons by the gap junctions	Li et al., 2019 [219]
	Cortical neuronal cells exposed to oxidant injury and MSCs n vitro	Co-culture	↑ Neuronal survival ↓ Oxidative injury	MSCs Mt transfer to damaged neuronal cells Role of Miro1 and TNT formation	Tseng et al., 2021 [157]
	Oxidative damage of hippocampal neurons (in vitro) and seizure mouse model	MSC-EVs (containing Mt)	↓ Hippocampal neuronal damage ↓ Sequelae of seizures in mice	↑ Antioxidant activity Nrf2 signaling pathway restoration ↓ ROS ↓ Mt dysfunction	Luo et al., 2021 [220]
	Rat TBI	MSCs or MSC- IL-10	MSCs-IL-10 > neuroprotection than MSCs alone	↑ Autophagy/mitophagy (NIX/BNIP3L), PINK-1 ↑ Cell survival ↓ Cell death ↓ Neuroinflammation	Maiti et al., 2019 [223]
	AD (ocadaic acid treated SH-SY5Y neuronal cells) n vitro	UC-MSC-CM	Cell protection Improved Mt function	EV and Mt transfer ↓ Expression of AD-related gene ↓ Mt oxidative stress	Zhang et al., 2020 [225]
	AD mouse model (repeated LPS-injections)	Regular and IL-6 primed MCSs	↓ episodic memory impairment Micro- and Microglial stimulation by primed MSCs	Improved Mt function in damaged brain cells ↓ Oxidative damage	Lykhmus et al., 2019 [226]
	Mouse MS	MSCs	↑ Myelinated areas	↑ Mt biogenesis ↑ Antioxidant levels	Shiri et al., 2021 [227]

Legend: ↑ increase; ↓ decrease; > greater.

## 5. Conclusions

The beneficial effects of MSCs in injured tissue and the underlying mechanisms of action have been the topic of intense preclinical research and clinical studies for decades. MSCs exert immunoregulatory effects supporting homeostasis and the resolution of injury/inflammation. They interact with the tissue microenvironment, modulate the functions of immune cells, secrete antimicrobial peptides, and improve mitochondrial functions in injured cells. Interaction between MSCs and injured tissues has bidirectional effects on mitochondria. Mt from damaged somatic cells are recognized by MSCs as danger-signaling organelles. These Mt are then engulfed and degraded by MSCs, leading to an increased expression of reparatory factors and boosting mitochondrial biogenesis in the MSCs. These effects increase the capacity of MSCs to donate their own Mt to injured cells. Healthy, donated Mt restore cell homeostasis in recipient cells by MQC regulation leading to improved function. This cascade of events was proven in both in vitro and in vivo models, and accumulating evidence suggests that it represents one of the key processes for cell and organ healing.

Clinical trials using MSC/MSC-EVs conducted so far show promising results. However, large-scale randomized trials with mechanistic studies conducted on patient samples are warranted. Mt transfer from MSC/MSC-EVs to injured cells is essential for MQC regulation and the re-establishment of Mt function. All these effects set the stage for cell recovery and the proper regulation of homeostasis, crucial events necessary for tissue restoration and the improvement of organ function.

## Figures and Tables

**Figure 2 ijms-24-15788-f002:**
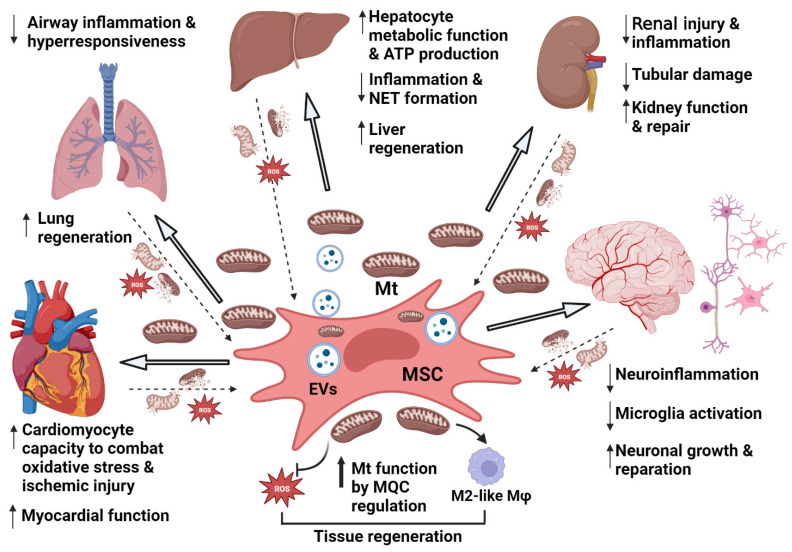
MSC/MSC-EV/Mt treatment of different injured organs induces Mt uptake by damaged cells and functional improvement of affected organs (Created with BioRender.com). Donation of healthy Mt by MSCs to injured donor cells of different organs corrects mitochondrial dysfunction of recipient cells (through MQC regulation) and activates metabolic and/or immunomodulatory pathways involved in restoration of tissue function. Oxidative stress is reduced, and Mϕ shifted to M2-like phenotype. On the other hand, Mt released from damaged or dying cells represent a key environmental factor that increases Mt drive, secretion of anti-inflammatory and pro-reparatory factors by MSCs. Legend: ↑ increase; ↓ decrease; gray arrows are pointing to the organs.

## Data Availability

Not applicable.

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
