# Peer review of "Therapeutic Effects of Mesenchymal Stromal Cells Require Mitochondrial Transfer and Quality Control"

_ijms, 2023, doi:10.3390/ijms242115788_

Round 1
Reviewer 1 Report
Comments and Suggestions for Authors
The authors Mukkala et al., present the review "Therapeutic Effects of Mesenchymal Stromal Cells Require Overall Mitochondrial Transfer and Quality Control". This is a well-constructed review and very pertinent to the area of understanding MSC's effect on MQC.
While this manuscript is very good and near publication quality there are a few minor corrections that will require addressing and comments are provided here for the author's consideration before acceptance.
- On page 2 line 57 acronym "Mt" is used first and not defined, please correct.
- On page 2 line 63 and 65 the links to clinical trial url includes in search terms in text. This may not be required as the reference list has the links of both reference 12 and 13
- unfortunately reference 13 in the bibliography is missing text, please correct.
- line 93 (Burt et al, 2019) is not in the numerical in-text referencing system used throughout the manuscript. please correct here and check the reference list as well.
Author Response
The authors Mukkala et al., present the review "Therapeutic Effects of Mesenchymal Stromal Cells Require Overall Mitochondrial Transfer and Quality Control". This is a well-constructed review and very pertinent to the area of understanding MSC's effect on MQC.
Response: Thank you for your opinion and comments!
While this manuscript is very good and near publication quality there are a few minor corrections that will require addressing and comments are provided here for the author's consideration before acceptance.
- On page 2 line 57 acronym "Mt" is used first and not defined, please correct.
Response: Done! Although the acronym “Mt” is introduced in the Abstract (line 18), reviewer is right, it should be also introduced in the text.
- On page 2 line 63 and 65 the links to clinical trial url includes in search terms in text. This may not be required as the reference list has the links of both reference 12 and 13
Response: Done!
- unfortunately reference 13 in the bibliography is missing text, please correct.
Response: Corrected! More references for clinical MSC studies are added as well!
- line 93 (Burt et al, 2019) is not in the numerical in-text referencing system used throughout the manuscript. please correct here and check the reference list as well.
Response: Done! Thanks a lot for noticing it. The description of the results from the Burt et al., 2019 paper continues in the next sentence, where the study is cited by number. Also, all references were double checked!

Reviewer 2 Report
Comments and Suggestions for Authors
Author Response
This article is a comprehensive review of the differentiation capacity of MSCs in reducing inflammation/injury, mainly by their effects on immunoregulatory function of immune cells, but also on transfer of mitochondria, and or mitochondrial fragments, from MSCs to other cells.
Although much of this work has been underway for years, the authors are careful to hold back on their therapeutic potential effects pending additional animal and then confirmatory clinical work. The most interesting aspect of this review is the suggestion that MSCs may enhance the mitochondrial quality control in immune cells or “injured tissue” that are infected by MSC mitochondria transferred via tunneling nanotubes, gap junctions, and microvesicles. The beneficial effects on MQC in MSCs, as in other cell types, looks to involve the usual pathways of biogenesis, fission/fusion, and mitophagy (PPARg/PGC1a, Parkin1, Mfn1,2, Drp1, PINK1, etc).
The illustrations are colorful and useful to summarize mechanisms.
Response: Thank you for your opinion and comments! Also, it is very helpful that you have listed some of crucial papers that should be added and improve our review paper.
What is not so clear is the mechanism of bi-directional, dynamic signaling between the MSCs and the targeted injured/inflamed cells. Is it only undifferentiated cells that have the capability for mitochondrial transfer? Are intact mitochondria transferred via gap junctions, nanotubes, of microvesicles, or are just mitochondrial fragments transferred to build new mitochondria in the recipient cells?
Response: Thank you for these interesting and important questions, they are answered below, one by one. Also, we clarified bi-directional mitochondrial transfer between MSCs and damaged cells (page 9, paragraph 3, lines 401-413) as well as between somatic cells (e.g. glial and injured neuronal cells; page 21, paragraph 2, lines 729-733).
1) It might be good to start with a discussion of what is a true “stem” cell and differentiate between other types (ESCs, HSCs, MSCs) and why MSCs, vs other undifferentiated cell types, are preferred by researchers in developing mitochondrial techniques to treat inflammatory diseases (e.g., see 10.1159/000290897).
Response: The relevant literature uses both terms, i.e. mesenchymal stem or stromal cells; and sometimes they are used interchangeably used. The International Society for Cell & Gene Therapy (ISCT®) (Viswanathan et al., 2019; https://doi.org/10.1016/j.jcyt.2019.08.002) published clarification of these terms. We have added new section (1.1. Definition and characteristics of MSCs) explaining also why MSCs are the most promising option in the field of cell-based therapies. The paper, link of which you kindly provided, is also cited in the section.
2) Are true stem cells more active in mitochondrial transfer than cells with some differentiation? That is could you use H9C2 cells for mitochondrial transfer in cardiac muscle disease, or HEK cells for kidney disease?
Response: We could not find direct comparison in the literature of mitochondrial transfer activity between true stem cells, MSCs or differentiated cells. As explained in section 1.1., apart from active Mt transfer to injured cells, MSCs are preferred for therapeutic use, because they are immunomodulatory and immuno-privileged cells (allowing for their allogeneic use), and because of relative simplicity of their isolation and culture.
3) Here are two more recent papers 10.3390/ijms24108848 10.1038/s41392-020-00440-z that the authors may want to include.
Response: We thank the reviewer for this suggestion. The papers are included in the new 2.3. subsection: “Intercellular mitochondrial transfer role in tissue homeostasis”, on page 7 (lines 276-297).
4) Please explain difference between whole mitochondrial transfer and mitochondrial component transfer and why one or the other is more beneficial.
Response: This is very important and intriguing question. We addressed this in the above-mentioned subsection 2.3. (pg. 7, lines 276-297). Clearly more research is needed to fully understand the benefits of transfer whole Mt and/or their components and have definitive answer to your question.
5) Please verify if only undifferentiated cells can transfer mitochondria to a differentiated somatic cell and why that be so.
Response: The short answer in no, as Mt could be transferred between somatic cells, as horizontal or vertical transfer. That was clarified in the section 2.3. (Intercellular mitochondrial transfer role in tissue homeostasis; page 7, lines 276-297). However, in case of MSC therapy, differentiation of MSCs into adipocytes, chondrocytes, osteoblasts etc. certainly is not favored. More clarification could be found on page 9, paragraph 3 (lines 401-413). Apart from potential differences in the efficiency of transfer, the practical advantage of stem cells as donors is also discussed, as mentioned above.
6) Unclear if transfer of mitochondria from MSCs is preferential for immune or inflammatory cells (i.e. macrophages) in clearing or repairing damaged cells (e.g., neurons, myocytes), or directly to damaged neurons/ myocytes.
Response: It is hard to say what would be preferred, but certainly, Mt from MSCs are transferred to both, inflammatory and damaged somatic cells. We refer back to the above discussion, and we acknowledge in text that future in-depth research is needed to systematically compare the efficiency and usefulness of various donor cells.
7) If MSC-induced mitochondrial transfer is so promising for practically every malady in all cell groups, how would treatment actually be carried out since MSCs would need to be delivered across a broad area of tissue damage, and the repair may be temporary?
Response: Obviously, this relevant question is the subject of current and future investigation. A number of facts addressing this point is succinctly reviewed in Section 1, focusing on the role of Mt transfer in MSC favorable effects. Thus:
MSCs could be given to the site of injury, e.g. in spinal cord (Hu et al., 2023; https://pubmed.ncbi.nlm.nih.gov/37342099/), intra-articularly (Al-Najar et al., 2017; https://pubmed.ncbi.nlm.nih.gov/29233163/), intraperitoneally (Izci et al., 2023; https://pubmed.ncbi.nlm.nih.gov/37814067/), intra-trachealy etc. However, intravenous MSC injection is also frequently used. Masterson et al., (2021; https://pubmed.ncbi.nlm.nih.gov/33664277/) used E. coli mouse pneumonia model and used intra-vital imaging of fluorescently labelled MSCs to determine their fate. The authors have found increased retention of MSCs in the pulmonary microvasculature in infected animals. Trapped MSCs deformed over time but retained therapeutic efficacy against pneumonia and appeared to release microvesicles. Similarly, Yarygin et al. (2021; https://pubmed.ncbi.nlm.nih.gov/34831220/) explored, in review paper, if the intra-arterially transplanted mesenchymal stem cells cross the blood–brain barrier (BBB) and could be effective in cell stroke therapy. The data they have collected showed that “some of the transplanted MSCs temporarily attach to the walls of the cerebral vessels and then return to the bloodstream or penetrate the BBB and either undergo homing in the perivascular space or penetrate deeper into the parenchyma. Transmigration across the BBB is not necessary for the induction of therapeutic effects, which can be incited through a paracrine mechanism even by cells located inside the blood vessels.” Zhu et al., (2023; https://pubmed.ncbi.nlm.nih.gov/37487541/) pointed out dynamic interplay between MSCs and the immune systems in the process of wound healing and outlined current insights of MSC ability to sense and modulate inflammation. It seems that MSCs are attracted by pro-inflammatory molecules and achieve their beneficial effects mainly by paracrine factors, secreted EVs and mitochondrial transfer, and immune-modulation.
8) Is the mechanism of transfer dependent on different factors, i.e., activation of one protein leads to nanotube tunnelling, another protein leads to vesicular transfer?
Response: Another very intriguing question for which current literature has limited answers. Some known proteins and pathways are mentioned in Section 4, however, future research is warranted to offer more in-depth understanding of these complicated mechanisms.
9) A deleterious effect of directed MSC-induced mitochondrial transfer could be enhanced stimulation of cancer cell development, rejection, or tissue degeneration; should this be mentioned?
Response: Yes, we agree, as Mt received from their surrounding cells could have important role in cancer survival, increased proliferation and invasion capabilities. At the same time, Mt could be used as powerful targets in cancer therapy. It is added at the end of Section 2.3. (page 7, lines 276-297).
Because this review comprehensively covers current and past research in field of MSC mitochondrial transfer, and its potential for treating various diseases in a well done fashion, the authors may respond to these comments in their revision, limited to the extent that the responses may be helpful to the readers of this article.
Response: Once again, thank you for all these important and intriguing questions.

Reviewer 3 Report
Comments and Suggestions for Authors
The authors presented a manuscript summarizing current knowledge on the mitochondrial transfer mediated by mesenchymal stem cells (MSCs). The publication is well-written and provides a comprehensive overview of the current knowledge on this topic. The authors have effectively presented the relevant details necessary for understanding.
However, I would like to suggest that the manuscript should also include a discussion on the possible adverse effects of MSC application. This addition would enhance the completeness of the review and provide a more balanced perspective on the subject.
Additionally, I have concerns about the quality of the figures. It appears that they may have been created using the free version of BioRender. Given this, there is a question whether they can be used in publications.
Author Response
mediated by mesenchymal stem cells (MSCs). The publication is well-written and provides a comprehensive overview of the current knowledge on this topic. The authors have effectively presented the relevant details necessary for understanding.
Response: Thank you for your opinion and comments!
However, I would like to suggest that the manuscript should also include a discussion on the possible adverse effects of MSC application. This addition would enhance the completeness of the review and provide a more balanced perspective on the subject.
Response: We agree with the Reviewer. Adverse effects of MSC application are now discussed in the manuscript (page 2, lines 61-72).
Additionally, I have concerns about the quality of the figures. It appears that they may have been created using the free version of BioRender. Given this, there is a question whether they can be used in publications.
Response: We have been using the professional version of BioRender. We have downloaded high resolution images for our Figures and these were included in the submission package separately. Licenses for publication of both Figures are also included in the submission package.
